# Smartphone scanning is a reliable and accurate alternative to contemporary residual limb measurement techniques

**Sam Walters**[1]*, **Benjamin Metcalfe**[2], **Martin Twiste**[3], **Elena Seminati**[4,5]☯, **Nicola Y. Bailey**[6]☯

1 Department of Mechanical Engineering, University of Bath, Bath, Somerset, United Kingdom, 2 Bath Institute for the Augmented Human, University of Bath, Bath, Somerset, United Kingdom, 3 School of Health and Society, University of Salford, Manchester, Greater Manchester, United Kingdom, 4 Department for Health, University of Bath, Bath, Somerset, United Kingdom, 5 Centre for the Analysis of Motion, Entertainment Research Applications (CAMERA), University of Bath, Bath, Somerset, United Kingdom, 6 Department of Engineering, King's College London, London, Greater London, United Kingdom

☯ These authors contributed equally to this work.
* sw2935@bath.ac.uk, samcwalters1@gmail.com

**Data Availability Statement:** Data are available at the University of Bath Data Archive, under the following DOI: https://doi.org/10.15125/BATH-01462.

## Abstract

Monitoring the volume and shape of residual limbs post-amputation is necessary to achieve optimal socket fit and determine overall limb health, yet contemporary clinical measurement techniques show high variance between measures. Three-dimensional scanning presents an opportunity for improved accuracy and reliability of residual limb measurements, however, three-dimensional scanners remain prohibitively expensive. A cost-effective alternative is the use of software that can utilise the photographs of modern smartphone cameras to create geometrically accurate scans. Whilst several studies have investigated the potential of privately developed photogrammetry algorithms for capturing residual limbs with clinical accuracy, none to the authors knowledge have explored commercially available software to do the same. Three applications were tested, namely Polycam, Luma, and Meshroom, to determine if they could produce clinically acceptable results. Scans of ten residual limbs were created using both smartphone technology and a reference structured-light scanner (Artec EVA), against which the validity and reliability of the resulting limb models were assessed using the Bland-Altman method and Intraclass Correlation Coefficient, respectively. Polycam and Luma achieved both Pearson Coefficients and Intraclass Correlation Coefficients of 0.999, and Coefficients of Variation of 1.1% and 1.4%, respectively. Volume reliability coefficients were 58.3 ml and 70.0 ml respectively for Polycam and Luma, whereas Meshroom failed to meet any of the criteria for clinical suitability, with a repeatability coefficient of 790.3 ml. Both Polycam and Luma exhibit sufficient accuracy and reliability to be considered for clinical volume measurements.

**Funding:** This project was funded by the Engineering and Physical Sciences Research Council [grant number: EP/W00717/1] through TIDAL Network Plus - Transformative Innovation in the Delivery of Assisted Living Products and Services. The funders had no role in study design, data collection and analysis, decision to publish, or preparation of the manuscript.

**Competing interests:** The authors have declared that no competing interests exist.

## Introduction

The socket is a critical part of a limb prosthesis, it is the interface between the rigid prosthesis and the residual limb. A consequence of poor socket fit is discomfort, which is the leading cause of abandonment across various types of prosthesis [1]. This problem can be eased by early adoption of a prosthesis post-amputation [2], but this contends with significant residual limb volume reduction due to chronic muscle atrophy and a decrease in post-surgical oedema [3, 4]. During this course to stability, a residuum may lose between 17–35% of its initial post-amputation volume [5], necessitating frequent socket revisions to ensure a snug fit. In a survey composed predominantly of people with established amputations, Pezzin et al. found that the surveyed group visit clinics nine times a year on average for such adjustments [6]. Once the residual limb has stabilised, there remain diurnal changes in volume (up to ±11%) caused by fluid transfer [3], necessitating the use of ply socks to compensate for the volume difference [7]. Monitoring the magnitude of these volume fluctuations is important to determining both the health of the limb and whether changes to the socket are required.

A common clinical method of deriving volume measurements is water submersion, where the residual limb is dipped into a cylinder of water and the displaced volume of water is measured. Starr et al. found the margin of error for this technique to be between 2.1–3.7% [8]. Alternatively, callipers and soft tape measures may be employed, using circumferential measurements to determine the volume [9]. These measurements deviate from water submersion measurements by ±8.1% [10]. A promising addition to the methods of clinical measurements is digital scanning, which provides simple measurement, sharing and storage of captured limb data, enabling easy comparisons between current and previous assessments. Scanning mediums can vary significantly both in terms of the fidelity of data and the cost of acquiring such data. Two of the most accurate and reliable means are Computed Tomograghy (CT) and Magnetic Resonance Imaging (MRI). These technologies are uniquely capable of acquiring internal data, such as the distribution of bone and soft tissue, which a prosthetist would otherwise have to determine qualitatively through palpation. Safari et al. explored MRIs potential as a means of measuring residual limb volume by comparing the outcomes of the hands-on and hands-off methods of Patellar Tendon-Bearing (PTB) socket fabrication [11]. However, the high cost and demand for these machines in other critical fields make them impractical for common clinical practice outside of research endeavours.

Structured-light scanning captures external surfaces by projecting a series of calibrated light patterns onto an object, and constructing geometric features based on how the light patterns deform around the surface [12]. These scans can be completed quickly and to a high degree of accuracy, with multiple studies supporting their capacity for accurately capturing residual limb volume [13–16]. Commercial structured-light scanners are capable of delivering results within 0.5% of the mean and exceeding the clinical threshold value for an ICC (Interquartile Correlation Coefficient) of 0.90 [16]. However, state of the art scanners remain considerable investments for clinics, especially when considering the current cost challenges facing healthcare providers such as the National Health Service (NHS) in the United Kingdom. This has prompted research into more accessible scanners within the hobbyist prince range, such as the iSense. This scanner retailed for £395 and uses similar principles as structured-light scanners, by projecting infrared light patterns whilst leveraging the camera of an iPhone or iPad. Armitage et al. found that although the iSense has adequate intra-rater reliability, the inter-rater reliability and range of error were substandard, with results of 0.85 and 3–6%, respectively [17]. In further tests by Dickinson et. al, the iSense was compared against the Sense and Omega scanners when scanning both plaster-casts and in-vivo residuums, where it performed the worst of the three across all metrics [18]. Alternatively, it is possible to perform scans with a device the

vast majority of clinicians already use in their daily clinical workflow, and that's smartphones [19, 20].

The use of smartphone cameras to capture patient data has become increasingly widespread in clinical practice, with many modern smartphone cameras matching or outperforming digital cameras in photograph quality [21]. High quality photographs are the core ingredient to one of the most fundamental of scanning technologies: photogrammetry. With roots in the early 1900s, where it was used for terrestrial land surveying [22], in recent years it has been employed for the fast and inexpensive creation of digital assets [23]. The principle of photogrammetry is the relative positioning of photographs detailing an objects surface from various angles and elevations, from which the external surface of the object can be inferred. Several studies have examined the use of both digital and smartphone cameras to capture static residual limb models, with which photogrammetry has been shown to yield digital models with degrees of accuracy as high as 99% of the scanned volume [24–26]. However, the suitability of photogrammetry for use on live patients remains debatable, with a previous study by Carbrera et al. demonstrating the difficulty of acquiring high-quality models with an independently developed photogrammetry algorithm [27]. In contrast, the authors have not found any examples in literature examining the many commercially available applications and software that allow users to convert photographs into digital models. As such, the question of whether commercial applications can produce residuum scans suitable for clinical measurements at minimal cost is investigated in this paper.

## Materials and methods

### Preliminary investigation

A preliminary search was conducted into applications that enable users to provide smartphone photographs for conversion to a 3D scan. The application criteria examined were the accuracy of the generated mesh and the ease with which the mesh could be created. Most attention was paid to smartphone applications, because they kept the user-experience as simple as possible by performing both photo capture and processing within the same application. An Apple iPhone 12 (Apple, 2020) was chosen as the smartphone with which to capture all photos used in the study, equipped with a 4.2mm f/1 dual wide camera. An iPhone was chosen because they have the highest proportion of users with 29% market share (followed by Samsung with 24% at time of writing) [28]. Beyond iPhone applications several desktop applications were explored, but Meshroom was the only accessible desktop application found to approach the established criteria, being an open-source application with a simple user-interface. The considered applications can be found in Table 1.

**Table 1. Table detailing the apps considered for this investigation.**

| Scanning Application | Version | Price per Month (£) | Scan Method | Platform | Passes? |
|---|---|---|---|---|---|
| 3DScannerApp* | 2.0.17 | 0 | Photogrammetry | iOS | No |
| Luma | 0.9.9 | 0 | NeRFs | iOS, Web | Yes |
| Meshroom | 2023.1.0 | 0 | Photogrammetry | Desktop | Yes |
| Metascan | 2.9.5 | 5.99 | Photogrammetry | iOS | No |
| Polycam | 3.2.15 | 14.99 | Photogrammetry | iOS, Android, Web, Google Play | Yes |
| ScandyPro* | 2.1.1 | 5.49 | Photogrammetry | iOS | No |
| Scaniverse | 2.1.8 | 0 | Photogrammetry | iOS | No |

* App requires the use of the front-facing camera to utilise the *true–depth* feature of the iPhone to improve the resolution of captured geometries.

Simple qualitative assessments were used for the initial filtering of suitable scanning applications, by using visual assessment of mesh and texture data produced from a variety of household objects. These objects were chosen for their unique geometrical features that would test the limits of the scanning applications in question, and establish their abilities beyond the comparatively simple geometry of a residual limb. Two such examples of scanning objects used were a coffee mug for testing through-hole and blind-hole generation, and a leafy plant for thin element generation. Each of these geometrical features offer opportunities for different kinds of mesh distortion to reveal itself that may otherwise be difficult to identify when scanning simple residual limbs. The quality of meshes compared to one-another was visually evident, reducing application choices to Polycam (Polycam Inc, 2020) [29], Luma (Luma AI Inc, 2022) [30], and Meshroom (AliceVision, 2018) [31]. Polycam is the most popular 3D capture software on the Apple App store [32], accessible to generation 6 iPhones onward and Android phones. At the time of writing the subscription cost for Polycam is £14.99 a month. Luma is a free application that uses NeRFs (Neural Radiance Fields) rather than photogrammetry [33], leveraging neural networks to generate a point cloud, that can be converted into a mesh. Both Polycam and Luma have websites to which photos can be uploaded and meshes generated, which were also evaluated to determine their validity for users without access to an iPhone. Meshroom is free and in contrast to Polycam and Luma, performs photogrammetry locally rather than using online cloud servers.

## Recruitment

Inclusion criteria for the study allowed for any level of major amputation, this being any amputation below wrist or foot disarticulation respectfully. This would allow trends between different amputation levels to be identified, which could inform different guidance on best scanning practices between them. Exclusion criteria necessitated participants be over the age of 18, and for their residuum being scanned to be at least 1 year into maturity. A sample size of seven participants with ten residual limbs between them were recruited, estimating an Inter-correlation Coefficient (ICC) higher than 0.95 (estimated error 5%, number of trials three per condition, confidence level 95%) [15]. Post-hoc analysis revealed the power of this study to be in excess of 0.99 across all applications tested, strongly suggesting a sample size of ten residual limbs is sufficient for providing statistically significant results. Participants were informed of all the experimental activities prior to the study taking place, and reinformed of these activities prior to the study commencing following confirmation of their written and verbal consent. The study received a favourable ethical opinion by the Research Ethics Approval Committee for Health (REACH) at the University of Bath (reference 23–031). Data collection took place between 11 September 2023 and 30 November 2023, and the data were kept on a secure University server and anonymised.

## Data collection

The Artec EVA structured-light scanner (Artec Group, Luxembourg) [34] was used as the criterion, having previously been shown by Seminati et al. to provide both accurate and reliable residuum scan data, within 1.4% of the criterion volume [14]. With the exception of those generated by Artec and Polycam, all applications produced scan models of abitrary dimensions, necessitating corrective scaling for a valid comparison against the criterion. To address this issue, a reference object was used during each scan which could be digitally measured by the operator to determine an appropriate scaling factor. This was done by dividing the true value of 60 mm, $x_r$, by the distance measured between the mesh prongs, $x_c$, as detailed in Fig 1. It is important to note that the use of manual scaling with respect to a reference object may

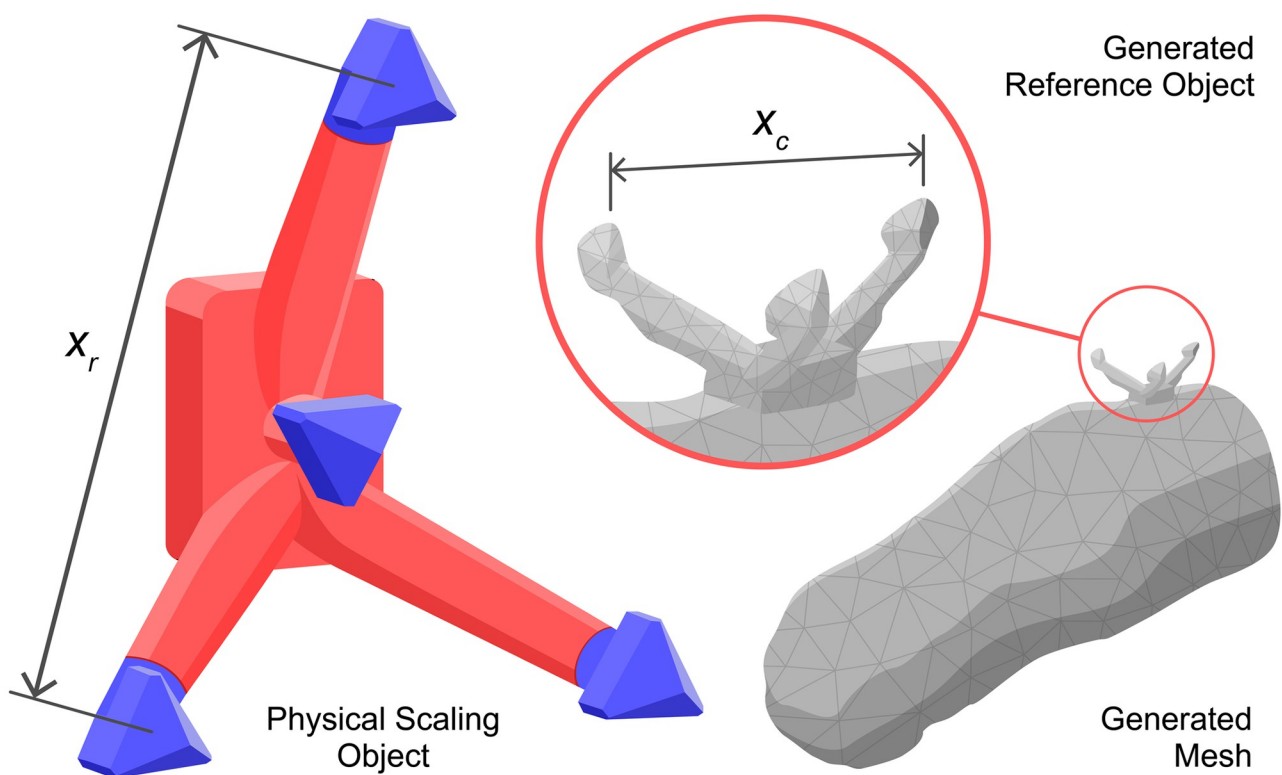

**Fig 1. Reference object with known dimensions, $x_r$ = 60 mm, and a digital scan with a corresponding measurement, $x_c$.** The scale is determined using $x_r/x_c$.

introduce a significant amount of human error, potentially clouding insights into the inherent accuracy of each of the tested scanning applications. Therefore, the study will record two datasets, namely *naive* and *optimal* scanning sets. The naive scan set uses solely the scaling object to scale the scan, whereas the optimal scan set have their scale corrected for any deviation from the Artec criterion scans in post-processing.

Due to scheduling conflicts, in vivo scanning of participants was performed between two different environments, so a brief quantitative assessment of the performance of each application in each environment was conducted. A plaster cast model of transtibial residuum was used, pictured in Fig 2a, and the difference in results between environments was found to be negligible. Consistent lighting was maintained within each environment and between participants. Additional environments and the resulting performance of the scanning applications in each can be found in S1 Table.

At least ten minutes prior to scanning of in vivo scanning, the participant doffed their prosthesis, because swelling of up to 6% post-doffing can occur within an 8-minute period [3, 9, 35]. The reference object was attached to the anterior side of the limb on a bony prominence if present, and fiducial markers of 10 mm diameter were dispersed on the limb's surface to aid tracking and scan alignment. The participant was asked to remain motionless during each scan and to position their limb in a way they found comfortable, as exemplified in Fig 2b. Photographs of the residuum were taken by the same operator approximately 0.5 m from the subject, however this distance decreased to within 0.3 m when capturing photographs below the residuum. The standard iPhone 'Camera' application was used for, Polycam Web, Luma Web and Meshroom, whereas the photographs for Polycam and Luma were captured within their

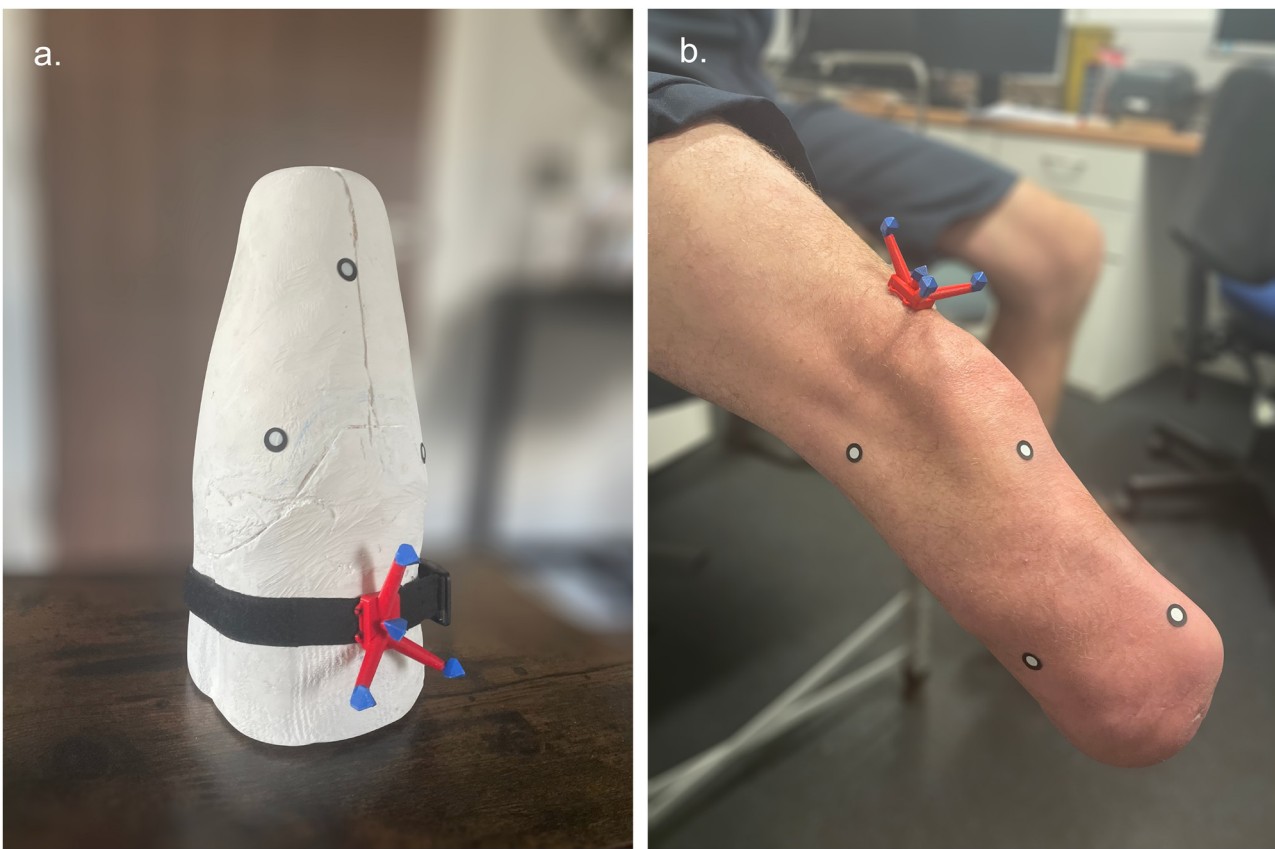

**Fig 2. Scanning setup for transtibial model and limb.** (a) Transtibial model used for evaluating the performance of each application between environments. (b) Transtibial limb of exemplar participant during scanning.

respective applications. The capture procedures were identical between applications, with the only significant difference being whether the photos were captured using the iPhone's default camera app (Luma Web, Polycam Web, Meshroom) or captured in-app (polycam, Luma). Consecutive frames overlapped by a minimum of 70% at multiple angles and levels of elevation [36], concluding after 2–4 minutes once the whole surface of the limb had been photographed (80–150 photographs). This variation is a result of the difference in sizes of the residuum and how it was positioned. A small case-study revealed the quality of scans appear to plateau after 75 photographs, as demonstrated in S1 and S2 Figs, where one of each major amputation level was examined. The capture process was repeated three times, by the same operator for each application, enabling analysis of the reliability of each application on a limb-by-limb basis.

## Data processing

Polycam and Luma smartphone data were uploaded to their respective cloud platforms for mesh creation. Photographs captured with the iPhone camera application were uploaded to the web versions of Polycam and Luma, and imported into the Meshroom desktop software. The time for each application to generate a mesh was between five to ten minutes with the exception of Meshroom, which took between two to three hours per scan. Once the models had been generated, each were exported as stereolithography (STL) files or object (OBJ) files, in the highest-quality format for each application. Post processing to remove background

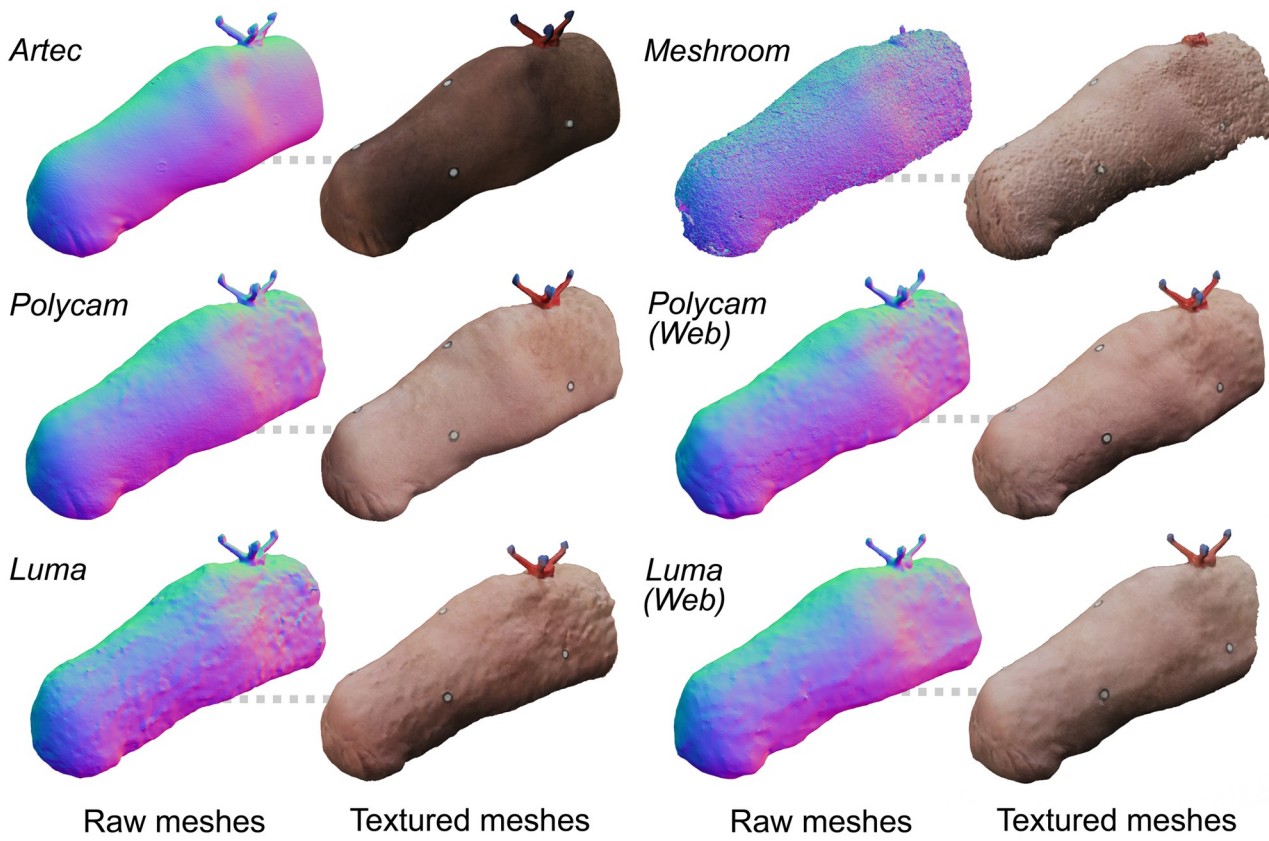

**Fig 3. Processed scans of exemplar participant's transtibial limb for each application, showing surface quality and output texture.**

geometry and floating artifacts, and perform corrective scaling using the virtual reference object was conducted in Blender (Blender Foundation, 2022) [37]. The mesh size and density were not altered for each application, with the exception of merging overlapping vertices. Fig 3 shows the meshes for each application post-processing.

All measurements were conducted in Artec Studio 12 Professional (Artec Europe, 2017) [34], in addition to hole patching which filled holes in meshes. Hole meshing was only necessary where insufficient data was collected, which was almost exclusively in low-light regions on the posterior side of scanned residuums. All scans were aligned relative to the criterion scans using the Artec Studio 12 rigid alignment algorithm, which minimises the error between geometric features. A base plane to serve as the proximal reference limit was established perpendicular to the limb's axis just beneath the reference object, or beneath the knee/elbow joint for transtibial and transradial limbs. The distance between this plane and the end of the limb was measured, and ten sections were created along its length, as shown in Fig 4.

## Statistical analysis

The three repeated scans from the criterion were used as the ground truth for each participant to calculate the validity of the different measurement systems. Reliability was assessed in terms of intra-rater reliability (repeatability), since the sole operator captured the data three times for each participant. Three main variables were assessed: the perimeter along each of the ten sections, the cross-sectional area (CSA), and the volume of the residual limb, measured from the

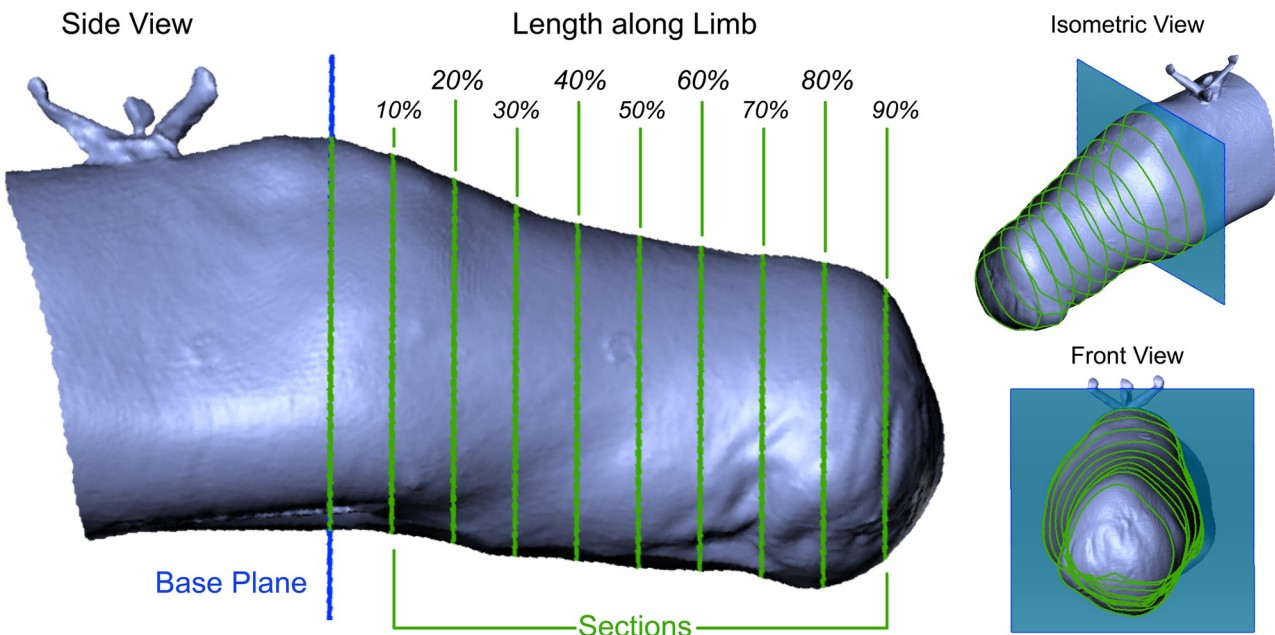

**Fig 4. Sectioning of an Artec EVA scan of an exemplar transtibial participant in Artec Studio 12.** Left: Lateral view of sectioned limb; Top right: Perspective view; Bottom right: Transverse-plane view.

base-plane. The raw measurements (mm, mm$^2$ and ml, respectively) were converted into standardised measurements expressed as a percentage for comparison purposes. The naive set and optimal set were analysed separately, and the presented data refers to the optimal dataset unless otherwise stated.

**Validity.**    The Bland-Altman method [38] was used to determine the limits of agreement between the criterion scans and the applications. The bias of each application show whether the application generally under- or over-predicted the volume and by how much. Bland-Altman analysis also yields the Pearson Correlation Coefficient and the Coefficient of Variation (CV). This analysis was conducted for the volume, and for each section of the residual limbs for each application, yielding results across the length of the limb for CSA and perimeter. To assess geometrical differences in the residual limb model, the Root Mean Square Error (RMSE) was calculated, by comparing each scan to the criterion scan that required the least post-processing. Within a designated search distance, each plane of the criterion mesh was sampled along it's normal to determine how far the closest surface from it is. The search distance was limited to 15 mm to minimise internal intersections of surface geometry, derived from taking the smallest measured perimeter across all sections to get a minimum diameter of 45 mm, and further divided by three as a conservative margin of error. The surface quality was assessed visually by determining vertex density and distribution across meshes, in conjunction with using heatmaps generated from RMSE data to highlight surface abnormalities that would compromise its utility for measurement purposes.

**Reliability.**    Consecutive pairwise analysis was conducted on each variable (volume, perimeters, and CSAs) for each application between each repeated measurement. From this the changes mean (the mean difference between the repeated scan results), typical error of the measurements (TEMs; standard deviation of the difference scores of all three repeated scans in the group divided by $\sqrt{2}$), and overall repeatability ($1.96\sqrt{2}$TEMs) were derived [38]. The Intraclass Correlation Coefficient (ICC) was calculated, with a 95% Confidence Interval (CI)

**Table 2. Information and measurements of participants scanned.**

| Participant | Limb Class | Mass (kg) | Height (m) | Years since Amputation | Age (years) | Amputation Cause |
|---|---|---|---|---|---|---|
| A | TF | 103 | 1.87 | 1 | 43 | Orthopaedic surgery failure |
| B | TT | 67.1 | 1.64 | 13 | 56 | Infection |
| C | TT | 92 | 1.89 | 7 | 60 | Trauma |
| D | KD | 140 | 1.72 | 6 | 56 | Trauma |
| E | TR, TH, 2xTF* | 65 | 1.67 | 10 | 43 | Infection |
| F | TT | 72.9 | 1.61 | 6 | 73 | Orthopaedic surgery failure |
| G | TT | 95.2 | 1.85 | 45 | 65 | Trauma |

* Participant E accounted for four of the ten limbs scanned. Limb classes included are transtibial (TT), knee-disarticulation (KD), transfemoral (TF), transradial (TR) and transhumeral (TH).

[39, 40], defining how closely the measurements from each application aligns with each other when used across different subjects. In line with clinical practice, an ICC of 0.90 was taken to be an acceptable threshold for these measurements [41].

## Results

The ten residual limbs scanned across seven participants, are listed in Table 2. There were five distinct levels of amputation covered, with transtibial amputation accounting for four of the ten, followed by transfemoral which accounted for three. Lower limb amputations accounted for eight of the ten limbs investigated, aligning with the typical prevalence of major lower-limb amputations over major upper-limb amputations [42]. Infection accounted for half the amputations, with the next most prevalent cause being trauma. Limb volumes ranged between 550 ml–2530 ml, measured using the Artec 12 Studio volume measurement tool. The average participant was 56.6 years old (SD = 11.0), living with an amputation for 12.6 years (SD = 14.8).

### Validity

As shown in Table 3, all applications except for Meshroom achieved a Pearson Coefficient for volume measurements greater than 0.99, surpassing clinical guidelines of 0.90 [43]. Polycam and Luma perform comparably to one-another by remaining within 1.1% and 1.4% respectively for their CV. All applications showed a tendency to under-predict the criterion volume, with web applications doing this to a greater extent than their smartphone-based counterparts. Polycam under-predicted volumes by an average of 20.7 ml, and Luma by an average of 7.1 ml, or by 2.9% and 1.0% of the overall volume respectively. This trend remains consistent across

**Table 3. Validity measurements for each application.**

| Scanning Method | Bias | | Limits of Agreement | | CV (%) | RMSE (mm) | Pearson |
|---|---|---|---|---|---|---|---|
| | Raw(ml) | Standardised (%) | Raw (ml) | Standardised (%) | | | |
| Polycam | -20.7 (−37.2, −4.2) | -2.9 (−5.3, −0.6) | 86.4 | 12.2 | 1.1 | 1.99 | 0.999 |
| Polycam(Web) | -50.1 (-91.8, -8.5) | -7.1 (-13.0, -1.2) | 218.5 | 30.9 | 2.6 | 2.31 | 0.989 |
| Luma | -7.1 (-23.0, 8.8) | -1.0 (-3.2, 1.2) | 81.9 | 11.5 | 1.4 | 2.36 | 0.999 |
| Luma(Web) | -6.5 (-25.5, 12.5) | -0.9 (-3.5, 1.7) | 96.1 | 13.3 | 1.5 | 2.04 | 0.998 |
| Meshroom | -145.7 (-270.9, -20.5) | -22.3 (-41.5, -3.1) | 594.5 | 91.1 | 10.3 | 2.50 | 0.886 |

Numbers enclosed in brackets indicate lower and upper confidence intervals respectively, where CI = 95%.

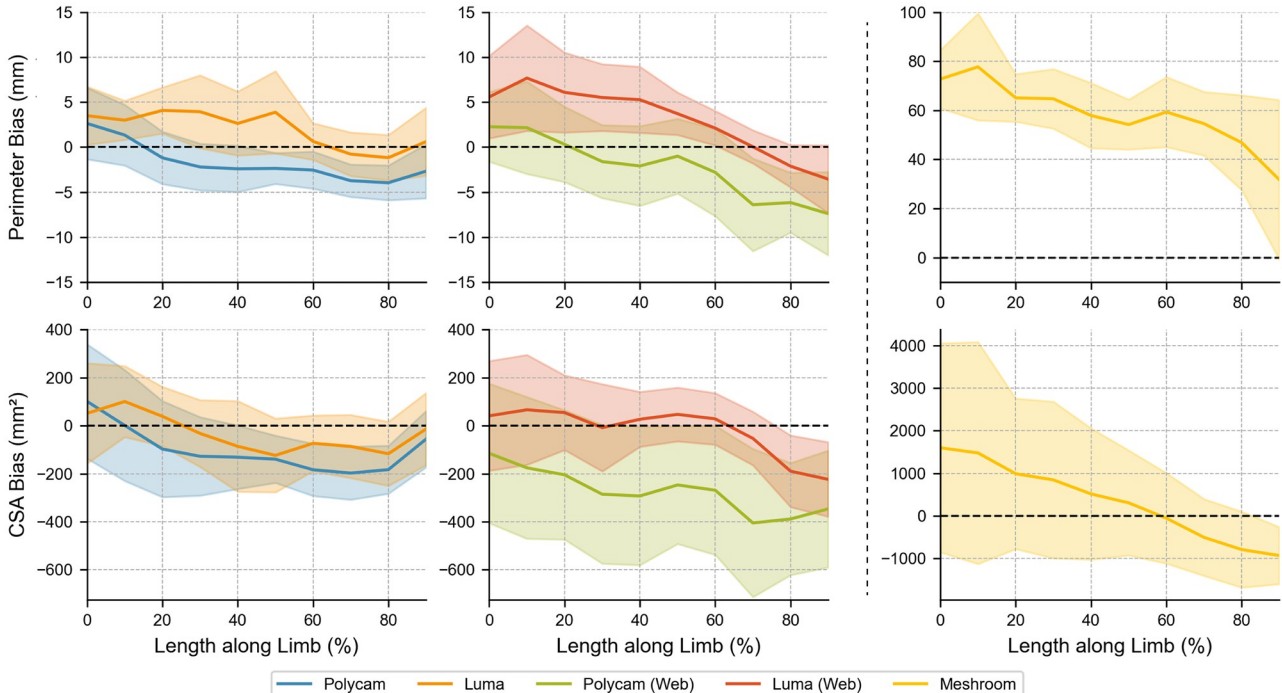

**Fig 5. Bias of perimeter and Cross Section Area (CSA) measurements.** Coloured bands indicate 95% confidence limits. Note that Meshroom necessitates different axis values to the other applications due to the significantly worse validity.

CSA measurements, however, Luma also showed an over-prediction of the perimeter in the first half of the limb, before dropping closer to the criterion in the latter half as shown in Fig 5. The standardised measurements exhibit an increase in confidence intervals closer to the ends of the residual limb, which is consistent across all applications. These visualisations can be found in S3 and S4 Figs of the Supporting Information, respectively.

Whereas Artec exhibited a strong consistency in vertex density, other applications experienced significantly more variance between repeated scans and participants, as demonstrated in Table 4. Fig 6 shows the distribution of mesh density across each application for an exemplar participant. Artec appears homogenous, whereas Polycam and Polycam Web exhibit higher mesh densities around the reference object and fiducial markers. The average RMSE for the criterion is only 1 mm, whereas Polycam and Luma achieved around double with 1.99 mm and 2.36 mm, respectively. The RMSE appears largely dependent on the amputation level of the residuum being scanned, as demonstrated in Fig 7. Luma exhibits a steeper increase in error than Polycam as the volume of the limb increases, although it should be noted that all methods, including the criterion, exhibit this upward trend to some extent.

Further, there is a clear deviation between the RMSE of the anterior and posterior of the limb. This is true for all limb classes, having posterior RMSEs on average 1.8 times greater than their respective anterior sides, reflecting the lower vertex-density in these regions. Fig 8a exemplifies the RMSE distribution of various scans from participant C. There was a significant difference observed in the surface quality of Polycam and Luma scans. Luma produced scans with greater surface roughness than Polycam, and Artec, frequently producing crevices in the posterior of the limb, whereas Polycam produced smooth surfaces closer to the criterion. Fig 8b shows an example of Participant A's scans, where the proximity of the limb to the torso

**Table 4. Mesh density across participant scans for each application.**

| Participant | Limb Class | Average No. of Vertices per cm$^2$ | | | | | |
|---|---|---|---|---|---|---|---|
| | | Artec | Polycam | Polycam (Web) | Luma | Luma (Web) | Meshroom |
| A | TF | 126.6 (0.6) | 6.9 (3.6) | 7.2 (1.7) | 41.8 (10.5) | 43.4 (11.8) | 184.7 (54.9) |
| B | TT | 127.5 (1.3) | 9.5 (2.8) | 9.8 (2.1) | N/A | 39.2 (36.7) | 261.7 (123.7) |
| C | TT | 127.9 (1.6) | 20.4 (4.8) | 24.6 (3.1) | 39.5 (7.3) | 33.6 (0.6) | 103.9 (43.6) |
| D | ED | 126.7 (0.6) | 10.0 (0.9) | 11.1 (0.3) | 33.8 (3.4) | 37.7 (5.9) | 207.0 (27.8) |
| E | TR | 128.3 (2.6) | 43.2 (10.2) | 36.6 (6.7) | 43.5 (1.7) | 50.2 (4.5) | 185.3 (26.7) |
| E | TH | 128.2 (1.8) | 65.2 (5.7) | 72.5 (3.1) | 61.5 (2.6) | 60.1 (1.1) | 379.1 (67.5) |
| E | TF1 | 121.4 (1.4) | 20.0 (1.1) | 22.6 (3.0) | 120.4 (21.5) | 146.4 (71.7) | 282.5 (13.4) |
| E | TF2 | 129.5 (0.3) | 19.5 (2.7) | 20.5 (1.7) | 219.3 (149.1) | 330.8 (23.2) | 330.5 (47.4) |
| F | TT | 129.3 (0.6) | 26.1 (3.7) | 22.2 (5.1) | 49.8 (1.7) | 46.1 (1.8) | 280.2 (35.7) |
| G | TT | 128.5 (2.6) | 33.7 (1.3) | 25.8 (8.6) | 57.7 (14.3) | 45.1 (7.1) | 135.5 (36.0) |

Vertex density was calculated by dividing the number of vertices in a mesh by the surface area of the mesh. The listed averages were achieved by taking the mean of the vertex densities for each repeated scan. Numbers in brackets indicate the standard deviation between these repeated scans.

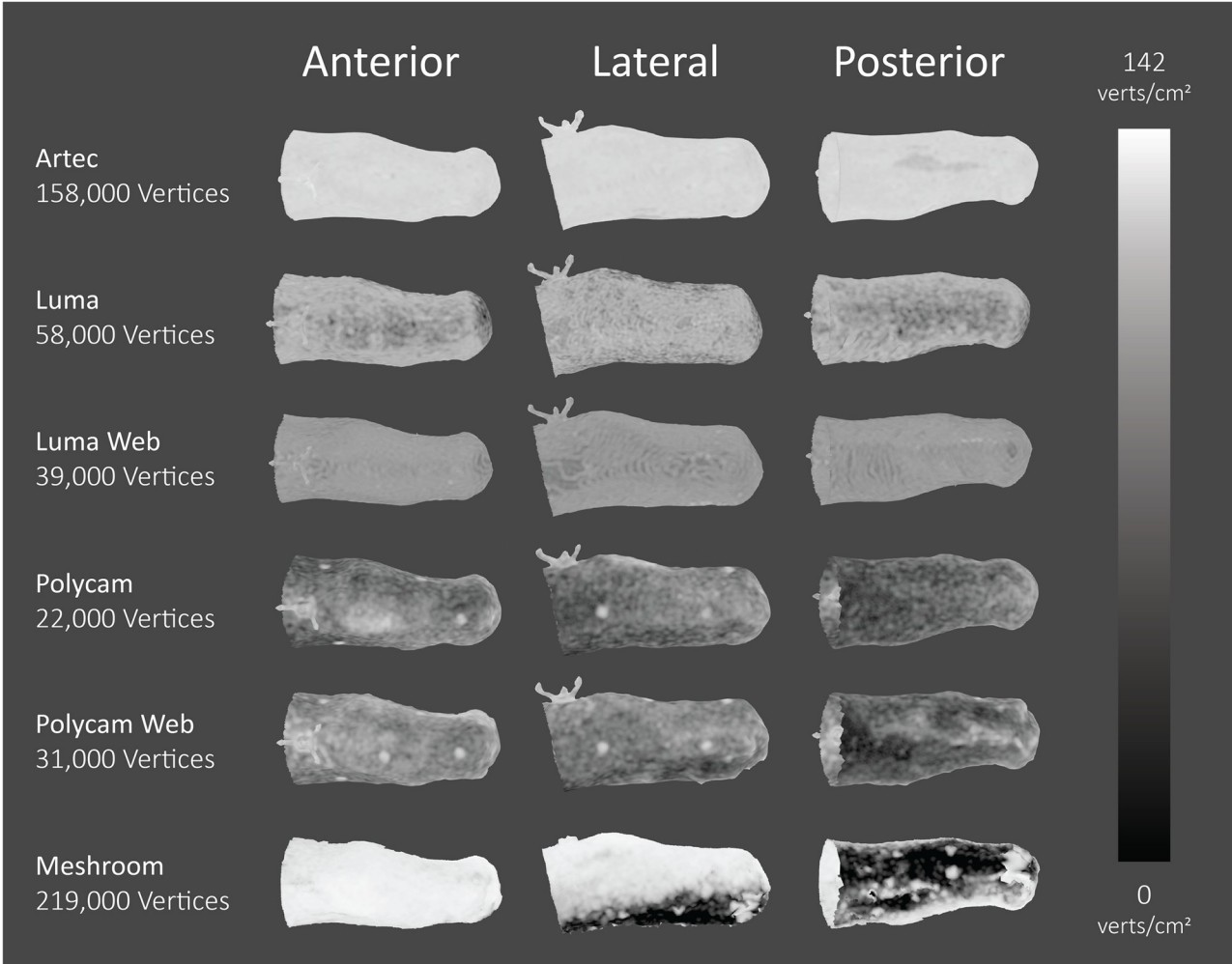

**Fig 6. Vertex-density maps of exemplar participant's transtibial limb for each application.** White areas indicate regions of high vertex density, whereas black areas indicate regions of low vertex density.

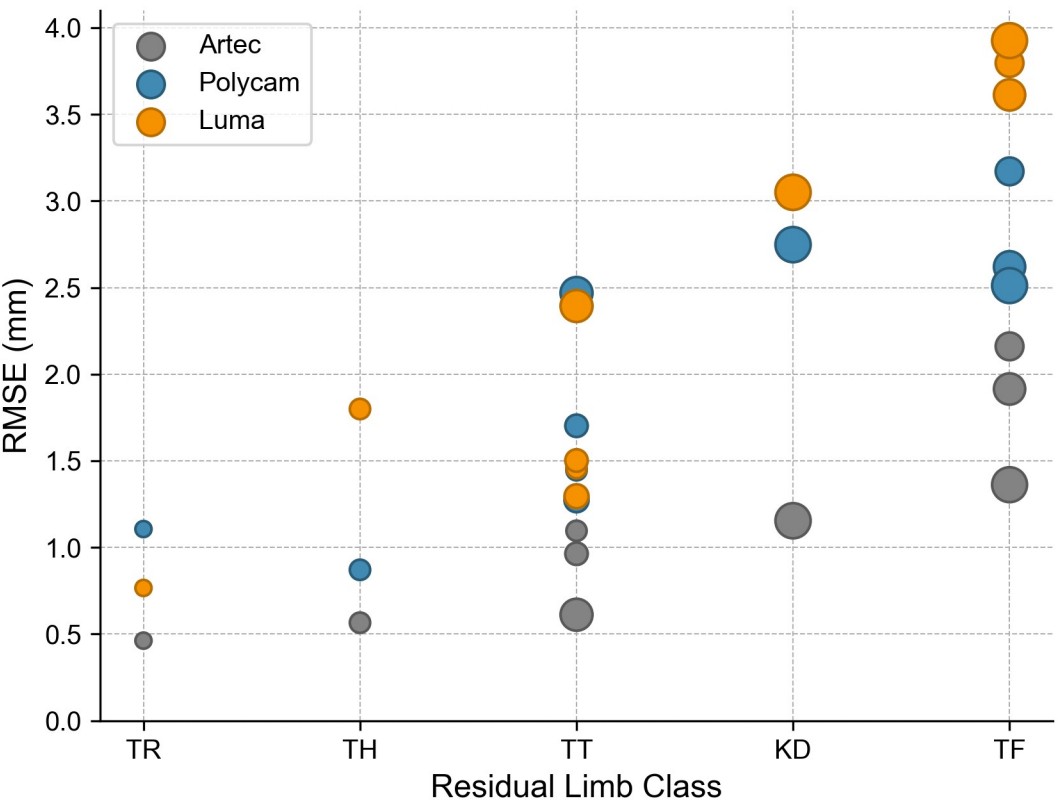

**Fig 7. RMSE of limbs according to limb class, for Artec, Polycam and Luma.** The size of the marker indicates the limb volumes relative to one another. Limb classes are transtibial (TT), knee-disarticulation (KD), transfemoral (TF), transradial (TR) and transhumeral (TH).

caused more shadow to fall on the inner-thigh region. Hence, the lateral side also suffered from surface irregularities typically seen on the limb posterior.

## Reliability

Table 5 shows that the ICC of all applications, except for Meshroom, exceed 0.90, indicating suitable reliability for clinical purposes [41]. Polycam and Luma perform within a typical error range of 5% for CSA measurements, but exceed this point as the length along the limb increases. Fig 9 shows Luma experiences significant volume differences around the centre of the residual limb, far exceeding Polycams error in this range, resulting in large confidence intervals. The overall repeatability for Polycam and Luma was found to be 58.3 ml and 70.0 ml, respectively. Their respective web applications experienced greater error in each circumstance, and Meshroom experienced error of an entire order of magnitude greater, as evidenced in Table 5.

## Impact of scaling

Fig 10 demonstrates the average difference in volume from the criterion for each application. Results show the volume difference is at least double for the naive set across all applications compared to optimal, and the same is true for their respective standard errors. Polycam and

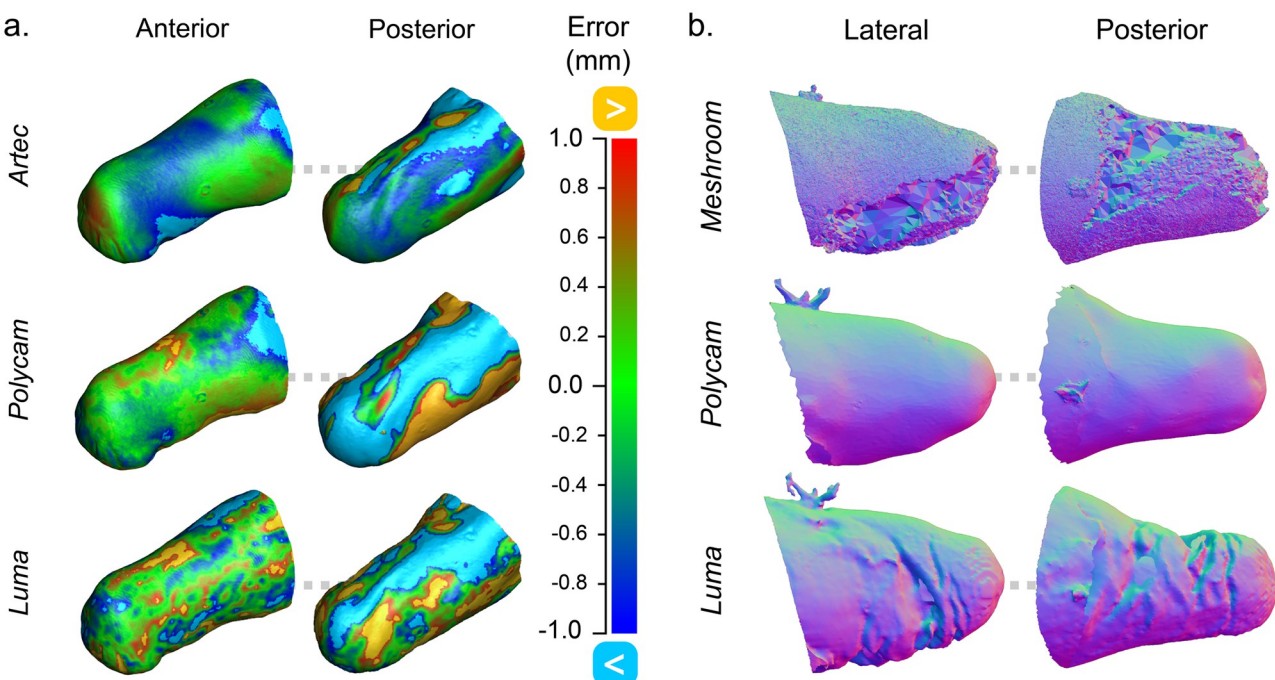

**Fig 8. Surface quality examination of residual limb meshes in the optimal set.** (a) RMSE values demonstrated on heatmaps of transradial limb of exemplar participant. (b) A participant's transfemoral limb scans, from a session wherein surface quality was poor.

Luma remain the best performing applications, with the naive set resulting in a difference of 4.2% and 4.4%, respectively.

## Discussion

Previous studies have investigated the use of portable tablets or smartphones for scanning residual limbs [17, 24, 25], however, these were limited to privately developed algorithms or hobbyist scanners that faced limited success. In this paper, the performance of commercial applications in producing residuum scans to assist in the monitoring of residual limb volume and shape using photographs captured with a smartphone has been assessed for the first time. Fernie and Holliday's findings show that a socket's fit remains comfortable within a 5% volume reduction and 2.5% volume gain [44]. These numbers are in line with the findings of Lilja

**Table 5. Reliability measurements for each application.**

| Scanning Method | Change in Mean | | TEM | | Repeatability | | ICC |
|---|---|---|---|---|---|---|---|
| | Raw (ml) | Standardised (%) | Raw (ml) | Standardised (%) | Raw (ml) | Standardised (%) | |
| Artec | -1.7 (-15.5, 12.0) | -0.2 (-2.1, 1.6) | 12.8 (9.4, 21.6) | 1.8 (1.3, 2.9) | 35.6 | 4.9 | 1.000 |
| Polycam | 2.5 (-18.4, 23.3) | 0.4 (-2.6, 3.3) | 21.0 (15.6, 34.7) | 3.0 (2.2, 4.9) | 58.3 | 8.2 | 0.999 |
| Polycam(Web) | 11.8 (-73.9, 97.4) | 1.7 (-11.0, 14.4) | 84.7 (62.1, 133.7) | 12.5 (9.1, 19.7) | 234.7 | 34.5 | 0.989 |
| Luma | -2.4 (-28.5, 23.8) | -0.3 (-4.0, 3.3) | 25.3 (18.6, 41.3) | 3.5 (2.6, 5.7) | 70.0 | 9.7 | 0.999 |
| Luma(Web) | -3.0 (-43.4, 37.5) | -0.4 (-5.8, 5.0) | 38.6 (28.5, 65.5) | 5.2 (3.8, 8.8) | 106.9 | 14.3 | 0.998 |
| Meshroom | -86.2 (-397.4, 224.9) | -15.5 (-71.9, 40.8) | 285.1 (202.7, 509.1) | 45.6 (32.4, 81.4) | 790.3 | 126.4 | 0.841 |

Numbers enclosed in brackets indicate lower and upper confidence intervals respectively, where CI = 95%.

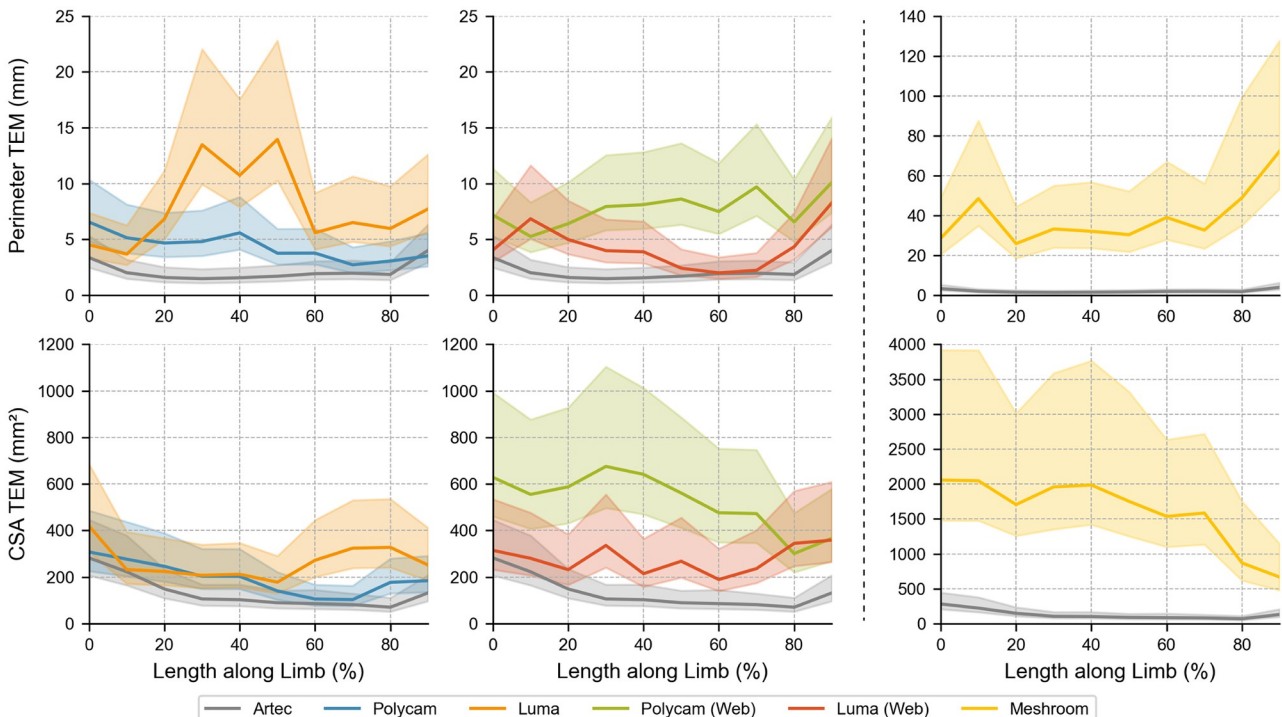

**Fig 9. Typical Error Measurements (TEM) for the perimeter and Cross Section Area (CSA) measurements across all applications.** Coloured bands indicate 95% confidence limits. Note that axis limits are different for Meshroom.

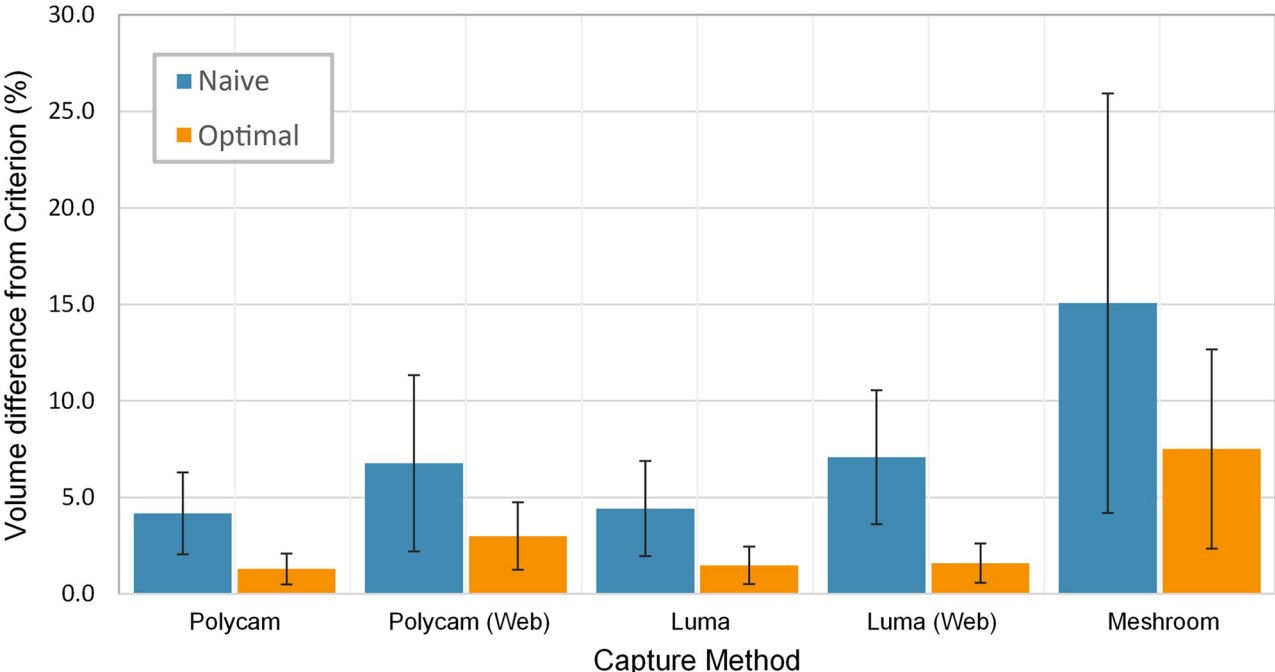

**Fig 10. Difference between criterion and the *naive* & *optimal* scaling sets, expressed as a percentage of the criterion volume.** The error bars represent the standard error between measurements within each set.

et al., who documented that a change in residuum volume of 5% is comparable to a single terry sock [5]. Whilst both Polycam and Luma had low bias at 20.7 ml and 7.1 ml, sitting comfortably within the aforementioned range, Meshroom exhibits a volume difference of 145.7 ml ($\sim$ 22% of the total volume) and therefore cannot be considered suitable for clinical use. Despite Meshrooms acceptable performance in the preliminary investigation, the application suffers significantly when the movement and shadows inherent to in-vivo scanning are introduced. Polycam *Web* and Luma *Web* both perform to an acceptable but lesser degree than their smartphone-based counterparts, with Polycam *Web* exhibiting the greater disparity in results. This could be the result of each smartphone-based application employing different camera settings than the iPhones default Camera application to improve the outcome of scans, hence the production of rougher surfaces and lower quality meshing of shaded regions.

The extent of shape differences between the anterior and posterior of the residual limb has been previously documented by Safari et al., when measuring the difference between scans and plaster cast models [11]. The dominant factor was determined to be loose soft tissue, which also plays a significant role in our investigation. Soft tissue can hang differently between scans depending on multiple factors, including the participant posture, the angle of their limb, and the level of muscle activation in the residuum at time of scanning. This problem may be exacerbated in clinical practice where the time between scans could be a matter of weeks rather than minutes. This is partly the reason why higher amputation levels exhibit greater RMSE values compared to limbs of lower amputation levels with comparable volume. Transfemoral and transhumeral residuums store a greater proportion of soft tissue, hence the potential for a shift in the soft structure increases. This problem is made more apparent as the relative mass of the participant increases, due to the greater proportion of soft tissue. The other dominant factor causing these discrepancies is lighting in the scanning environment. The seated position each participant assumed allowed stability and resembled the way gravity would typically pull on the soft tissue of the limb. However, this means the posterior remains in shadow when lifted horizontally, despite efforts to ensure uniform lighting. This difficulty is again exacerbated with lower-limb amputations where the limb remains closer to the floor, making uniform coverage tricky. These factors suggest that smartphone scanning may be best suited to patients with upper-limb amputations, and that improvements to the scan collection process may be necessary when dealing with lower-limb amputations.

In terms of reliability, most applications exceeded the 0.90 thresholds for clinically relevant reliability in ICCs, with Polycam and Luma achieving clinically acceptable results, while the worst performance was again for Meshroom (ICC = 0.84). Polycam and Luma performed similarly to a common 3D scanner currently used in clinical practice (e.g., the OMEGA Scanner 3D), with repeatability coefficients of 58.3 ml and 70 ml) [15]. They can potentially be useful in the early stage post-amputation where changes in volume and shape are relatively large. Artec intra-rater reliability (repeatability) was found to be greater than previously recorded values in residual limb models (35.6 ml compared with 13.94 ml). It is reasonable to assume it is due to this investigation's scans being taken in vivo rather than using plaster cast models, thus introducing movement artifacts. This difference is well precedented by Commean et al. who found reduced reliability measuring in vivo compared to plaster casts [45]. However, the repeatability for Artec found in this study was lower compared with data reported from residual limbs (67.22 ml in Seminati et al. 2022). All applications, including the criterion, showed a tendency for the standardised error to increase towards the end of the limb, as can be seen in the Supplementary Information. The web-based applications exhibit this to a greater degree, for both perimeter and cross-sectional area. However, the difference in performance is stark when measuring perimeter, where Luma exhibits significantly more variation than Polycam.

There is a clear difference between the *naive* and *optimal* datasets. Under the circumstances of intended use outlined in this paper, the clinician will not have access to a criterion scan to compare the scan against, and as such must rely on the reference object to produce accurate dimensions. As previously demonstrated, the geometries themselves are capable of producing clinically applicable results, but only with a reliable method of minimising scaling error could such scans be implemented in clinical practice. The introduction of such methods to close the gap between naive and optimal datasets would allow these highly accessible means of acquiring accurate and reliable residuum scans to be swiftly incorporated into clinical practice with minimal investment.

## Limitations & future directions

The biggest limitation of this study is the need for accurate scaling by the operator to achieve clinically acceptable results. Without the use of LiDAR or equivalent hardware to automatically determine scan dimensions, only the reference geometry can be relied on to achieve the desired scale, where the validity becomes highly subject to human error. As such, it would be advisable to develop an automatic system that aligns the scan to a digital counterpart of the reference geometry, and applying the appropriate scaling operations to maximise alignment. In conjunction with this aim, studies should also prioritise streamlining or potentially automating the mesh adjustment and measurement stage. The current method requires expensive software and knowledge of computer-aided design which clinicians may not have. Automation of this pipeline could reduce human error and minimise the time spent processing each scan for the clinician. This could improve the consistency of the scaling process, one of the largest current sources of human error, as evidenced by Fig 10. If successful, the use-cases are not limited to just residual limbs, but could be expanded towards the development of other orthopaedic devices, such as for use with orthoses, splints and orthopaedic shoes.

Another limitation to this study was the need for hole-filling tools to complete meshes. This unavoidably introduces a degree of error, and although the approximated hole-filling geometry likely contributed to the greater RMSE values identified in low-light regions, they are not the sole cause. Instead, these holes are a symptom of poor lighting conditions and insufficient visual data capture from the posterior of the residuums scanned. Improving the lighting conditions under the posterior of the limb, for example by introducing a diffused light source to illuminate it from below, will likely reduce hole instances significantly.

The frequency with which new applications are released limits exhaustive investigation of available solutions. The medium of smartphone 3D scanning is young, and as previously discussed, modern smartphone cameras currently match or surpass the fidelity necessary for clinical use [21]. As such, it's the authors opinion that the majority of improvement in smartphone scanning capabilities will likely come from advancements in software rather than hardware. This has already been demonstrated by Luma, wherein the NeRFs used can reproduce residuum scans with a significantly smaller photo-set than those used by software employing typical photogrammetry methods. As such, continued monitoring of developments in smartphone scanning applications across iOS and Android platforms is recommended.

Future studies should focus on determining the differences in mesh quality between different operators when performing a residuum scan, particularly operators without prior scanning experience. As promising as the current results are, the operator became experienced with the best practices of each application and hence more variation in results may be expected from a layperson. Such an investigation could be accomplished with three operators who have each been trained for a short 30-minute period prior to in vivo scanning, and a control operator with extensive 3D scanning experience. If these studies yield results comparable to those

produced by this study, such that multiple novice operators may create clinically acceptable residual limb scans, it is feasible to suggest that smartphone scanning may not require a trained clinician to perform. This could allow people with residual limbs to have friends or family perform scans of their residual limbs in their own home, reducing clinic visits that may be unnecessary, and allowing clinicians to monitor a patient's residuum remotely.

Other studies may develop an independent application for smartphones, specifically catered towards capturing accurate and reliable residuum scans. A formalised and free smartphone resource such as this could benefit those in less economically developed countries, where standardised assessment of residual limbs is otherwise not available. Despite low and middle income countries harbouring 30–40 million of the worlds people with amputations, only 5% have access to prosthetic care [46]. For this population, which will only grow larger in the next decade [47], remote assessments of the residual limb could prove invaluable. Yet more beneficial for this population could be the use of such an application for the generation of digital sockets, that could be 3D-printed to provide personalised care, helping close the wide gap in access to prosthetic care between developing and developed countries. Although there is no replacement for a clinician's skills and expertise, mobile scanning could provide a new avenue for patients with little to no access to such fundamental care. The wide-spread creation of such digital residuum assets could theoretically be used for further research purposes. Not only do digital residuum scans facilitate the sharing of high-fidelity information between clinicians, but a collection of such scans could be used to construct an anonymous database for the purpose of research. It's a well-known problem in prosthetics research that acquiring appropriate sample sizes is difficult, due to the small size of the population with amputations, hence an open-source database of limb scans may go some way to alleviate these difficulties in particular research cases.

## Conclusion

Both Polycam and Luma have potential to be utilised within and outside of clinical practice for the measurement of residual limb volumes, with optimal captures matching criterion model volumes within 2.9% and 1.0% respectively. They can accommodate fluctuations and results can be stored and assessed by clinicians. The capture process is simple for each, although Polycam requires a paid subscription. Although both applications performed well, it is recognised that Polycam is likely the best current candidate for adoption, as it produces surface geometry with fewer defects in low-light areas than Luma, and it does not require rescaling, a significant source of human error. Polycams textured models can capture the surface details of the limb in adequate detail, such as scar tissue and blemishes, which could prove useful for remote analysis of the limb's health by clinicians. However, should a clinician or person with an amputation not have access to a smartphone that supports Luma or Polycam, it is recommended that the photo-set is uploaded to Luma *Web*. This is due to Luma *Web* producing similar results to Luma across the board, whereas Polycam *Web* consistently exhibited lower accuracy and reliability. The excellent surface quality of Polycam's meshes provide a promising opportunity for use in the manufacture of sockets with CAD/CAM technology, using the residual limb surface as a foundation.

## Supporting information

**S1 Table. Deviation between repeated measurements for applications across a range of environments.** Numbers enclosed in brackets indicate standard error measurements. Unfortunately the data for Polycam *Web* in the *Studio*, *Office* and *Outdoors* environments were lost. (PDF)

**S1 Fig. Graphs showing RMSE differences depending on size of photo-set.** Left: RMSE values across a transtibial (Participant C), transradial (Participant E), transhumeral (Participant E), and transfemoral (Participant A), for different photoset sizes. Sampled photo-set sizes include 25, 36, 50, 75, 100, 125, and 150 photographs. Right: Signed distances across the same limbs, and their maximum and minimum deviations from the criterion surface.
(PDF)

**S2 Fig. Effect of photoset size on quality of generated meshes.** A range of models of a transtibial residuum using the same photoset with varying numbers of photographs made available for the algorithm to use. As demonstrated, there is little difference in surface quality between 75 photographs and 150 photographs, but a significant difference between the other photoset sizes.
(PDF)

**S3 Fig. Standardised validity measurements.** Coloured bands indicate 95% confidence limits. Note that Meshroom necessitates different axis values to the other applications due to it's significantly worse validity.
(PDF)

**S4 Fig. Standardised reliability measurements.** Coloured bands indicate 95% confidence limits. Note that Meshroom necessitates different axis values to the other applications due to it's significantly worse validity.
(PDF)

# Acknowledgments

We would like to thank Elise Pegg for her invaluable guidance in the interpretation and presentation of the gathered data, Mike Fray for his help in designing the methodology technique, and Richard Bibb for his support throughout the project.

# Author Contributions

**Conceptualization:** Martin Twiste, Elena Seminati, Nicola Y. Bailey.

**Data curation:** Sam Walters, Elena Seminati, Nicola Y. Bailey.

**Formal analysis:** Sam Walters.

**Funding acquisition:** Elena Seminati, Nicola Y. Bailey.

**Investigation:** Sam Walters, Martin Twiste, Elena Seminati, Nicola Y. Bailey.

**Methodology:** Sam Walters, Benjamin Metcalfe, Elena Seminati, Nicola Y. Bailey.

**Project administration:** Elena Seminati, Nicola Y. Bailey.

**Resources:** Sam Walters, Benjamin Metcalfe, Martin Twiste, Elena Seminati, Nicola Y. Bailey.

**Software:** Sam Walters.

**Supervision:** Benjamin Metcalfe, Elena Seminati, Nicola Y. Bailey.

**Validation:** Sam Walters.

**Visualization:** Sam Walters.

**Writing – original draft:** Sam Walters.

**Writing – review & editing:** Sam Walters, Benjamin Metcalfe, Martin Twiste, Elena Seminati, Nicola Y. Bailey.

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
