## [Decision Letter · Decision Letter 0]

4 Sep 2024

PONE-D-24-07058Remote scanning of residual limbs using smartphone technologyPLOS ONE

Dear Dr. Walters,

Thank you for submitting your manuscript to PLOS ONE. After careful consideration, we feel that it has merit but does not fully meet PLOS ONE’s publication criteria as it currently stands. Therefore, we invite you to submit a revised version of the manuscript that addresses the points raised during the review process.

We look forward to receiving your revised manuscript.

Kind regards,

Fei Yan

Academic Editor

PLOS ONE

Journal Requirements:

   "This project was funded by the Engineering and Physical Sciences Research Council [grant number: EP/W00717/1] through TIDAL Network Plus - Transformative Innovation in the Delivery of Assisted Living Products and Services."

4. In the online submission form, you indicated that "Data is available upon request to the authors. Some data is unable to be shared, namely photos taken of participants, due to possible identifying information."

5. Please amend either the title on the online submission form (via Edit Submission) or the title in the manuscript so that they are identical.

Reviewers' comments:

Reviewer's Responses to Questions

**Comments to the Author**

1. Is the manuscript technically sound, and do the data support the conclusions?

Reviewer #1: Yes

Reviewer #2: Yes

2. Has the statistical analysis been performed appropriately and rigorously? 

Reviewer #1: Yes

Reviewer #2: Yes

3. Have the authors made all data underlying the findings in their manuscript fully available?

Reviewer #1: No

Reviewer #2: Yes

4. Is the manuscript presented in an intelligible fashion and written in standard English?

Reviewer #1: Yes

Reviewer #2: Yes

5. Review Comments to the Author

Reviewer #1: This study presents comparative data for the accuracy and reliability of 3D scanning conducted using smartphones and apps, in comparison to a clinically-used gold standard 3D scanning device.

This is a novel piece of work which will be useful to the rapidly growing digital prosthetics community, and I believe will be cited well should you accept it for publication. It is well supported by data and the authors have taken care over their data visualisation and images. The writing is acceptable but could be made more concise, as it currently includes quite a lot of qualitative, observational details which arguably should be explained in more quantitative detail, or could be cut (e.g. roughness, poor quality in low lighting). Ultimately though I thank the authors for an interesting study and would support its eventual publication following some relatively minor revisions, as follows.

General: this community typically avoids the terms 'stump' (prefer 'residual limb' or 'residuum') and 'amputee' (prefer person-centred term, 'person with amputation') - recommend changing throughout.

General: there are also fairly frequent uses of 'better' - could these be replaced throughout with more specific terms? See e.g. point 12 below as one example of a specific alternative, and 21 where it is unclear (Need a better term than better ;))

1 The title does not really represent the study, especially the short title; if focusing on 'remote scanning' I would argue the emphasis might be the community-based settings - specific conditions of scanning, like lighting, scanning operated by non-experts/patients/carers, etc. Something around "Accuracy and reliability of smartphone-based 3D scanning apps" might be more appropriate.

2 Abstract statement "however the question of whether the validity and reliability of the scans produced by these software could meet clinical standards has gone unaddressed." is a bit strong given the introduction references. Perhaps tone this down a little?

3 L7: 'oedema' ?

4 L33-7: the Sengeh reference is rather a departure to the concept- in that study MRI was not to capture shape, so reference to earlier review papers (Sanders & Fatone [9], https://doi.org/10.1155%2F2013%2F486146 using MRI to compare PCast and PTB designs) might be more appropriate.

5 L49-51: reference 16 also looked at the Sense scanner in models, and https://doi.org/10.1097/JPO.000000000000035 compared Omega, Sense and iSense scanners in living participants.

6 L75-77: references to support the idea of community-based scanning, and considerations specific to this, like environmental factors (lighting, heat) and non-expert operators, as presumably scanning/imaging might require the patient and their carer or family members instead of a trained clinician or community worker? This was pitched in

"Technologies to Enhance Quality and Access to Prosthetics & Orthotics: the importance of a multidisciplinary, user-centred approach in https://apps.who.int/iris/rest/bitstreams/1264724/retrieve#page=62)"

7: Recruitment details: structure as Inclusion and Exclusion criteria?

8: L126: pitch as a power calculation? Post-hoc would be acceptable!

9: L159: Were photography conditions standardised in any way? Lighting? Distance? etc.

10: L165-8: Accepting this about number of frames being 'indistinguishable', is there scope to consider testing the number of photos required as an independent variable, vs. accuracy of the result?

11: L186-8: need to manually scale seems to be a major limitation to the method / those devices, and I found the consideration of this as a variable a bit confusing. Could this be simplified, even removed?

12: Comment L192-4 on hole filling - could be mentioned in the Discussion as a limitation of these methods, or as part of the argument for investing in Sense or Artec devices?

13: Sorry if I missed it - what was the proximal reference limit / cutoff for volume measurement?

14: L263-4 are these averages?

15: L218-7: Report on the mesh size / vertex density generated by each method? And whether this was controlled in any way? Noise may well be related to fine mesh density - coarser mesh looks smoother, up to a point. There's an optimum. Or does this refer to all meshes of same size, so a true comparison?

16: Not sure what is meant by volatility?

17: L316: 'comfort threshold' might be reading a bit much into that limit - consider toning down?

18: L332 - add 'or muscle tension/activation' to how loose the soft tissue is?

19: L335-9: "higher or lower amputation levels" would be clearer. It's all the body. Assume you mean trans fem or hum vs trans tib or rad?

20: L378-9: unless you build the scanner and software, or have an open published tool (see e.g. doi:10.1109/ACCESS.2018.2843725, ought to be in Introduction too?) this is always going to be the case?

21: L388-9: Most smartphone cameras are now very powerful indeed, likely overkill for many applications. How do you envisage 'better' specifically?

22: Table 5: Check caption - no numbers in brackets...?

Reviewer #2: The "Preliminary Investigation" section presents the research rationale in a somewhat disorganized manner. I recommend including a flowchart to visually guide the reader through the research logic and results, which would clarify the investigative process.

Lines 103-104 mention that various household objects, including coffee cups and plants, were used to generate models through the applications, followed by an evaluation of scan quality. However, the selection criteria for these objects are unclear. What was the rationale behind choosing these particular items? Please provide justification for their selection.

In Lines 124-125, the paper references lower and upper limb amputees. Could you clarify whether these conditions influence the scanning outcomes? It would be beneficial to explore how such factors might affect the accuracy or quality of the scans.

Mobile scanning technology is already widely available, and the accuracy of these products has been well-documented. Please elaborate on the innovations of this study, particularly in addressing clinical problems and advancing scanning technology beyond existing solutions.

6. PLOS authors have the option to publish the peer review history of their article (what does this mean?). If published, this will include your full peer review and any attached files.

Reviewer #1: No

Reviewer #2: No

---

## [Author Response · Author response to Decision Letter 0]

16 Oct 2024

Dear Editor,

We would like to submit a revised version of the manuscript formerly entitled ‘Remote scanning of residual limbs using smartphone technology’ (Ref. No: PONE-D-24-07058), now entitled ‘Smartphone scanning is a reliable and accurate alternative to contemporary residual limb measurement techniques’. Please see our responses below, highlighted in red, to the reviewer’s comments.

We would like to take this opportunity to thank the reviewers for their helpful comments, which we hope you agree have significantly improved the manuscript. We look forward to your decision.

Response to Reviewer 1

We thank the reviewer for their insightful comments and have implemented proposed amendments into the manuscript where possible, and have performed additional investigations as prompted to validate our findings. With regards to the availability of data used in the study, we are currently in the process of uploading all relevant data captured in the study to the University of Bath Research Data Archive, which will be made publicly available.

General: This community typically avoids the terms 'stump' (prefer 'residual limb' or 'residuum') and 'amputee' (prefer person-centred term, 'person with amputation') - recommend changing throughout.

Thank you, we appreciate the importance of inclusive language and have updated the manuscript accordingly. All previous instances of ‘stump’ have been replaced with ‘residual limb’ or ‘residuum’, and all instances of ‘amputee’ have been replaced with ‘person with amputation’ or ‘person with limb difference’, depending upon the context.

General: There are also fairly frequent uses of 'better' - could these be replaced throughout with more specific terms? See e.g. point 12 below as one example of a specific alternative, and 21 where it is unclear (Need a better term than better ;))

The two instances of ‘better’ in the original manuscript have been replaced with more informative descriptors. The first was removed entirely during revision of the manuscript. The second instance (lines 431-432 of original manuscript) which previously said ‘… as it produces better surface geometry than Luma’ has been replace with ‘… as it produces fewer defects in low-light areas than Luma’.

1 The title does not really represent the study, especially the short title; if focusing on 'remote scanning' I would argue the emphasis might be the community-based settings - specific conditions of scanning, like lighting, scanning operated by non-experts/patients/carers, etc. Something around "Accuracy and reliability of smartphone-based 3D scanning apps" might be more appropriate. 

We thank the reviewer for highlighting this. We have reframed the article to align better with the evidence we presented and suggest remoted scanning as a potential future direction in the discussion section. This details the suggested methods that would be used to validate such a study, such as using multiple operators in various environments with a consistent limb being scanned to compare the differences between them. The new title is now: ‘Smartphone scanning is a reliable and accurate alternative to contemporary residual limb measurement techniques’.

2 Abstract statement "however the question of whether the validity and reliability of the scans produced by these software could meet clinical standards has gone unaddressed." is a bit strong given the introduction references. Perhaps tone this down a little?

The statement in question has been reframed to better reflect the current state of literature in the space: ’Whilst several studies have investigated the potential of privately developed photogrammetry algorithms for capturing residual limbs with clinical accuracy, none to the authors knowledge have explored commercially available software to do the same.’

3 L7: 'oedema' ?

This has been corrected to the appropriate ‘oedema’.

4 L33-7: the Sengeh reference is rather a departure to the concept- in that study MRI was not to capture shape, so reference to earlier review papers (Sanders & Fatone [9], https://doi.org/10.1155%2F2013%2F486146 using MRI to compare PCast and PTB designs) might be more appropriate.

The authors agree with the reviewers assessment of the citation, and as such it has been removed.

5 L49-51: reference 16 also looked at the Sense scanner in models, and https://doi.org/10.1097/JPO.000000000000035 compared Omega, Sense and iSense scanners in living participants.

We thank the reviewer for pointing out this article as an important piece of work on this topic. This study has been integrated into the Introduction section on page 3, referenced as such: ‘Safari et al. explored MRIs potential as a means of measuring residual limb volume by comparing the outcomes of the hands-on and hands-off methods of Patellar Tendon-Bearing (PTB) socket fabrication.’

6 L75-77: references to support the idea of community-based scanning, and considerations specific to this, like environmental factors (lighting, heat) and non-expert operators, as presumably scanning/imaging might require the patient and their carer or family members instead of a trained clinician or community worker? This was pitched in "Technologies to Enhance Quality and Access to Prosthetics & Orthotics: the importance of a multidisciplinary, user-centred approach in https://apps.who.int/iris/rest/bitstreams/1264724/retrieve#page=62)"

We thank the reviewer for another relevant paper suggestion, which has now been included. As raised in suggestion 1, the format of the paper has been restructured to more accurately reflect the evidence provided. The proposed work performed by Dickinson et. Al, has been included to reinforce the presented evidence regarding hobbyist scanners as follows: ‘In further tests by Dickinson et. al, the iSense was compared against the Sense and Omega scanners when scanning both plaster-casts and in-vivo residuums, where it performed the worst of the three across all metrics’.

7: Recruitment details: structure as Inclusion and Exclusion criteria?

The recruitment details have been modified to explicitly name the inclusion and exclusion criteria . This adjustment can be found in lines 110-115 (page 6): ‘Inclusion criteria for the study allowed for any level of major amputation, this being any amputation below wrist or foot disarticulation respectfully. This would allow trends between different amputation levels to be identified, which could inform different guidance on best scanning practices between them. Exclusion criteria necessitated participants be over the age of 18, and for their residuum being scanned to be at least 1 year into maturity.’

8: L126: pitch as a power calculation? Post-hoc would be acceptable!

We thank the reviewer for spotting this oversight and have completed a post-hoc power calculation and included the findings in the recruitment section, lines 118-120 (page 6). We found that the power of this study exceeded 0.99 for all applications tested, including Meshroom. The exact calculation we used can be found below using G*Power (v3.1.9.7), and effect size (ICC) taken from Table 4 of the original manuscript. 

9: L159: Were photography conditions standardised in any way? Lighting? Distance? etc.

We have commented in more detail that the subject was photographed from 0.3-0.5 m away, subject to the region of the limb being photographed. It is now explicitly noted that the differences in lighting conditions between the two environments is negligible to the overall study outcomes in lines 143-148 (page 7), stating ‘Due to scheduling conflicts, in vivo scanning of participants was performed between two different environments, so a brief quantitative assessment of the performance of each application in each environment was conducted. A plaster cast model of transtibial residuum was used, pictured in Fig. 2a, and the difference in results between environments was found to be negligible. Consistent lighting was maintained within each environment and between participants’. With regards to the distance, it is later stated in lines 156 - 158 (pages 7-8) that ‘Photographs of the residuum were taken by the same operator approximately 0.5 m from the subject, however this distance decreased to within 0.3 m when capturing photographs below the residuum’. 

10: L165-8: Accepting this about number of frames being 'indistinguishable', is there scope to consider testing the number of photos required as an independent variable, vs. accuracy of the result?

We thank the reviewer for this comment, and we conducted additional testing of the existing dataset to validate this statement. We took the largest photosets for each kind of residual limb in the investigation, namely a transtibial, transfemoral, transradial, and transhumeral limb. These were split into separate photosets with quantities of 25, 36, 50, 75, 100, 125, and 150 (where possible). Each of these photosets were then fed back into the Polycam PG and Luma PG algorithms, and the resulting limb scans were compared against a criterion. They were each aligned in Artec Studio and an RMSE analysis was conducted, as was conducted in the main study. This analysis revealed that the size of the photoset had a significant impact upon the accuracy of the scan output, however the difference in RMSE appears to plateau between 75-150 photos in the dataset, supporting our previous assertions. It was interesting to note that the size of the photoset had a significantly greater effect upon Polycam than it did luma. Whilst the size of the photoset did show a small positive correlation with the accuracy of the scan for Luma, the relationship was far smaller than we had previously assumed. This may be due to the incorporation of neural networks in Luma’s design that facilitates more missing angles, as opposed to Polycams more traditional photogrammetry approach. This information has been added to lines 163 – 166 (page 8) and to the supplementary material for interested readers, under Supporting Figure 1 (S1 Fig) . Alongside these graphs, a visualisation of the generated models is provided as a visual indicator of the quality of the mesh as the photoset used to generate it is increased, also presented in the supplementary material under Supporting Figure 2 (S2 Fig ).

11: L186-8: need to manually scale seems to be a major limitation to the method / those devices, and I found the consideration of this as a variable a bit confusing. Could this be simplified, even removed?

We agree that the need for manual scaling is a limitation of the investigated applications. However, we believe the consideration of this variable is important to the paper, and as such have better established the reasoning for including both scaling sets earlier in the paper. This is clarified in lines 133 – 137 (page 7) where these sets are introduced and justified: ‘It is important to note that the use of manual scaling with respect to a reference object may introduce a significant amount of human error, potentially clouding insights into the inherent accuracy of each of the tested scanning applications. Therefore, the study will record two datasets, namely naive and optimal scanning sets.’ The primary focus on the optimal scaling set is reinforced again later in lines 201 – 202 (page 9), where it is said: ‘The naive set and optimal set were analysed separately, and the presented data refers to the optimal dataset unless otherwise stated.’

12: Comment L192-4 on hole filling - could be mentioned in the Discussion as a limitation of these methods, or as part of the argument for investing in Sense or Artec devices?

We appreciate the reviewers concerns relating to access to hole-filling tools and how this may be a limitation in the process outlined in the study. To remedy this, we have mentioned discussed the creation of holed meshed further in the Limitations and Discussions section, outlining the likely causes of such holes, their effect, and how they may be remedied in future studies. Lines 380 – 387 (page 17) read: ‘Another limitation to this study was the need for hole-filling tools to complete meshes. This unavoidably introduces a degree of error, and although the approximated hole-filling geometry likely contributed to the greater RMSE values identified in low-light regions, they are not the sole cause. Instead, these holes are a symptom of poor lighting conditions and insufficient visual data capture from the posterior of the residuums scanned. Improving the lighting conditions under the posterior of the limb, for example by introducing a diffused light source to illuminate it from below, will likely reduce hole instances significantly.’

13: Sorry if I missed it - what was the proximal reference limit / cutoff for volume measurement?

We have added some clarifying statements to where we describe how the residuum is measured to make it clearer, under the Data Processing section, lines 190 – 194 (page 9): ‘A base plane to serve as the proximal reference limit was established perpendicular to the limb’s axis just beneath the reference object, or beneath the knee/elbow joint for transtibial and transradial limbs. The distance between this plane and the end of the limb was measured, and ten sections were created along its length…’. 

14: L263-4 are these averages?

Yes, and the text has been altered to make this explicit.

15: L218-7: Report on the mesh size / vertex density generated by each method? And whether this was controlled in any way? Noise may well be related to fine mesh density - coarser mesh looks smoother, up to a point. There's an optimum. Or does this refer to all meshes of same size, so a true comparison?

We thank the reviewer for this suggestion and have implemented it into the paper. Average vertex density for each mesh was determined by dividing the number of vertices in a mesh by the surface area of the mesh. Further analysis was conducted by producing heatmaps in Blender of the vertex density of each mesh; whilst these don’t yield quantitative data, they provide a useful insight into the vertex distribution typical for each application. These findings have been condensed into Fig. 6, a new figure now included in the study. The vertex density is discussed in further detail and used as supplementary evidence for our findings on the variation in RMSE between the anterior and posterior of the residuum. 

16: Not sure what is meant by volatility?

‘Volatility’ has been changed to a more appropriate ‘volume differences’ on lines 284 – 285 (page 13).

17: L316: 'comfort threshold' might be reading a bit much into that limit - consider toning down?

‘Comfort threshold’ has been altered to a more nuanced discussion based firmly within the findings of the study presented by Lilja et al., which can be found on lines 304 – 305 (page 14): ‘These numbers are in line with the findings of Lilja et al., who documented that a change in residuum volume of 5% is comparable to a single terry sock.’

18: L332 - add 'or muscle tension/activation' to how loose the soft tissue is?

Changed accordingly.

19: L335-9: "higher or lower amputation levels" would be clearer. It's all the body. Assume you mean trans fem or hum vs trans tib or rad?

The passage has been updated to reflect this, and other instances of the previous explanation have been updated accordingly.

20: L378-9: unless you build the scanner and software, or have an open published tool (see e.g. doi:10.1109/ACCESS.2018.2843725, ought to be in Introduction too?) this is always going to be the case?

We agree, and this point has been removed and replaced with a discussion of the problems with scaling, as previously touched on by the reviewer.

21: L388-9: Most smartphone cameras are now very powerful indeed, likely overkill for many applications. How do you envisage 'better' specifically?

It’s been previously noted that smartphone cameras are currently meeting or surpassing the necessary quality for use in clinical environments, hence it may be more suitable to suggest that improvements in the outcomes of smartphone scanners will come from innovations in the software and algorithms used to create such scans, rather than the photographs themselves. This has been discussed and reflected on in detail in the Limitations and Future Directions section.

22: Table 5: Ch

---

## [Decision Letter · Decision Letter 1]

28 Oct 2024

Smartphone scanning is a reliable and accurate alternative to contemporary residual limb measurement techniques.

PONE-D-24-07058R1

Dear Dr. Walters,

We’re pleased to inform you that your manuscript has been judged scientifically suitable for publication and will be formally accepted for publication once it meets all outstanding technical requirements.

Kind regards,

Fei Yan

Academic Editor

PLOS ONE

Additional Editor Comments (optional):

Reviewers' comments:

Reviewer's Responses to Questions

**Comments to the Author**

1. If the authors have adequately addressed your comments raised in a previous round of review and you feel that this manuscript is now acceptable for publication, you may indicate that here to bypass the “Comments to the Author” section, enter your conflict of interest statement in the “Confidential to Editor” section, and submit your "Accept" recommendation.

Reviewer #1: All comments have been addressed

Reviewer #2: All comments have been addressed

2. Is the manuscript technically sound, and do the data support the conclusions?

Reviewer #1: Yes

Reviewer #2: Yes

3. Has the statistical analysis been performed appropriately and rigorously? 

Reviewer #1: Yes

Reviewer #2: Yes

4. Have the authors made all data underlying the findings in their manuscript fully available?

Reviewer #1: Yes

Reviewer #2: Yes

5. Is the manuscript presented in an intelligible fashion and written in standard English?

Reviewer #1: Yes

Reviewer #2: Yes

6. Review Comments to the Author

Reviewer #1: Thanks for such thorough consideration of my comments and congratulations on a great study. I and happy to recommend publication

Reviewer #2: Authors has already answered all questions raised and made some modifications to the original manuscript. It is recommended to be accepted.

7. PLOS authors have the option to publish the peer review history of their article (what does this mean?). If published, this will include your full peer review and any attached files.

Reviewer #1: No

Reviewer #2: No

---

## [Editor Report · Acceptance letter]

15 Nov 2024

PONE-D-24-07058R1 

PLOS ONE

Dear Dr. Walters, 

I'm pleased to inform you that your manuscript has been deemed suitable for publication in PLOS ONE. Congratulations! Your manuscript is now being handed over to our production team.

Kind regards, 

on behalf of

Dr. Fei Yan 

Academic Editor

PLOS ONE